# Effect of Morphine Administration on Social and Non-Social Play Behaviour in Calves

**DOI:** 10.3390/ani9020056

**Published:** 2019-02-12

**Authors:** Mhairi Sutherland, Gemma Worth, Catherine Cameron, Else Verbeek

**Affiliations:** 1AgResearch Ltd., Ruakura Research Centre, Hamilton 3240, New Zealand; gem.worth@gmail.com (G.W.); Catherine.Cameron2@agresearch.co.nz (C.C.); 2Swedish University of Agriculature, Department of Animal Environment and Health, SE-75007 Uppsala, Sweden; else.verbeek@slu.se

**Keywords:** opioids, play, welfare

## Abstract

**Simple Summary:**

Play can be used as an indicator of welfare in animals, because animals play more when all their basic needs are met. Opioids have a modulatory effect on social play behaviour in rodents and primates, however little is known regarding the central mechanisms involved in play behaviour in ruminants. In ruminants, we need to know more about what factors influence play behaviour, to determine which elements of play may more accurately be used as indicators of positive welfare. Therefore, the objective of this study was to evaluate the effect of morphine on social and non-social play behaviour in calves. In an arena test, morphine administration increased the performance of social play events but had no effect on locomotor play in calves. Similar to research in rodents and primates, morphine administration appears to increase social but not non-social elements of play in calves, suggesting that increased social play may be more indicative of a positive affective state.

**Abstract:**

The objective of this study was to evaluate the effect of morphine on social and non-social play behaviour in calves. Twelve calves experienced four treatments in a cross over 2 × 2 factorial design: Calves received an intravenous injection of morphine or saline 10 min prior to being tested individually or in pairs in an arena for 20 min. Play behaviour was continuously recorded in the arena test. Lying times were recorded in the home pen. Cortisol concentrations were measured before and after testing. In the arena test, calves given morphine tended to perform more social play events than calves given saline, however, morphine administration had no effect on locomotor play. Calves given morphine spent less time lying than calves given saline during the first 4 h after returning to the home pen. Cortisol concentrations were suppressed in calves given morphine. Administration of morphine appeared to increase social play but had no effect on locomotor play in calves. This study highlights the importance of investigating different aspects of play behaviour in animals as some may be more indicative of a positive affective state than others. More studies investigating the effects of morphine on play are needed to confirm the results found in this study.

## 1. Introduction

Play behaviour in calves is reduced when conditions are suboptimal, such as in response to low milk allowance, weaning [1] and after dehorning [2]. Because play is reduced when animals are exposed to negative stimuli or environmental challenges, play has been identified as a potential indicator of welfare in animals [3]. Play, and in particular social play, is also a pleasurable and rewarding experience for young animals [4,5]. Play also serves an important biological function through shaping the development of behavioural flexibility as well as appropriate social and cognitive behaviours [6]. Therefore, play could also potentially be used as an indicator of good welfare. It is now widely accepted that it is not only important that animals do not suffer, but that they also have the opportunity to experience positive emotional states [7]. In order to ensure good animal welfare, we need to be able to assess positive experiences in animals.

The opioid system in the brain has been linked to positive emotional states [8]. For example, endogenous opioids have been connected with reward processing and stimulation of the µ receptor is the primary mechanism by which opioids enhance reward processes and can increase both the incentive to seek food and the rewarding effects of food [9]. The opioid system also plays an important role in social bonding and affiliative behaviours [9,10]. Research in rats has shown that endogenous opioids are released in several brain regions during social play [5,11]. Conversely, opioid antagonists (e.g., naloxone) can reduce social play behaviour in rodents [12]. These studies strongly suggest that endogenous opioids have an overall modulatory effect on social play behaviour in rats. Moreover, morphine facilitated social but not non-social play behaviour in primates [13], suggesting that non-social play may not be modulated by the endogenous opioid system. Therefore, more research is needed to understand the relationship between opioids and different types of play and to investigate the role of opioids in play behaviour in other species.

The opioid system also plays a role in regulating the hypothalamus-pituitary-adrenal (HPA) axis. It has been shown that morphine administration suppresses the milking machine-induced increase in cortisol in dairy cows [14]. Morphine administration also suppresses corticotropin-releasing hormone (CRH)-induced increases in plasma cortisol concentrations in sheep [15]. Isolation stress-induced increases in cortisol are also attenuated by morphine administration in sheep [16], suggesting that the opioid system also plays a role in modulating the HPA axis response during emotional distress in ruminants. However, it is not known whether morphine also suppresses cortisol concentrations during positive experiences such as play.

The objective of this study was to evaluate the effect of morphine on social and non-social play behaviour in calves. We hypothesised that morphine administration enhanced social play in dairy calves without affecting non-social locomotor play and would lead to suppressed cortisol concentrations.

## 2. Materials and Methods 

### 2.1. Animals, Housing and Design

This study was conducted between April and May at the AgResearch Dairy Research farm, South Waikato (175° 18 00′E longitude, −38° 03 00′S latitude), New Zealand. All procedures involving animals were approved by the AgResearch Ruakura Animal Ethics Committee (protocol no. 12883) under the New Zealand Animal Welfare act 1999.

Twenty-four Friesian bull calves were used in this study. Calves were sourced from a commercial farm where they were separated from their dams within 24 h of birth and transported to the farm’s calf rearing facility. Calves were kept in covered pens with floors covered in stones prior to being transported to the AgResearch Dairy Research farm at approximately 6 days of age. The calf rearing facility on the AgResearch Dairy Research farm had solid dirt floors and walls on all four sides. The walls were either solid or covered with shade cloth. The floor of the pens were covered with river stones (Mangatangi River Rock Ltd., Auckland, New Zealand) with an approximate diameter of 40 to 60 mm. The stones were laid at a depth of approximately 30 cm and had not been previously used at the beginning of the study. The purpose of rearing calves on river stones was to increase their motivation to play. Previous studies have shown that calves reared on stones play significantly more in an arena test than calves reared on wood shavings [17]. On arrival, the calves were allocated to one of six experimental pens (n = 4 calves/pen, 2 experimental and 2 play partners) balanced for age and body weight. Calves remained in these pens for the duration of the study. Pens were constructed from wooden panels and a steel, paneled gate, that allowed auditory, visual, olfactory and some tactile contact between animals and provided a space allowance of 1.5 m^2^/calf. All pens had plastic troughs for water and feed and these were attached to the side of the pen.

Calves were fed 2.5 L of colostrum twice a day at 07:30 and 16:30 for the first 4 days after birth. Thereafter, the equivalent amount of milk replacer was offered (Topcalf, Fonterra Ltd, Auckland, New Zealand) using a five teat milk feeder (Stallion Plastic Ltd., Palmerston North, New Zealand) which was removed after each feeding. Additionally, calves were given ad libitum access to commercial calf meal (Gusto, PGG Wrightson Grain, Te Awamutu, New Zealand). Water was provided ad libitum.

### 2.2. Experimental Design

Calves were approximately 6 weeks of age (60.1 ± 13.4 kg body weight, mean ± SD) at the start of the study. Twelve calves were used as experimental animals receiving the pharmacological intervention (morphine or saline) prior to testing and the other twelve were used as ‘play’ partners during testing (n = 4 calves/pen, 2 experimental and 2 play partners).

The 12 experimental calves experienced all four treatments in a cross over 2 × 2 factorial design: Morphine (MOR) or saline (SAL) administered intravenously 10 min prior to being tested individually (IND) or in pairs (SOC) with a pen-mate in an arena test for 20 min. Therefore, calves either received an injection of saline (SAL, n = 12) or morphine (MOR, n = 12), and were tested individually (IND, n = 12) or in pairs (SOC, n = 12). Saline or morphine sulfate (0.5 mg/kg, Hospira, Inc., Lase Forest, IL, USA) was administered intravenously into the jugular vein 10 min prior to entering the arena test. Two calves per treatment were tested per trial day. The dosage of morphine used was based on previously published research evaluating affective state in sheep [18]. Calves were given a 3 day recovery period between consecutive treatments. Calves were weighed the morning of treatments.

#### 2.2.1. Behaviour 

Calves propensity to play was tested either individually or in pairs in a play arena test. The arena measured 3.0 × 15 m^2^ and the floor of the arena was covered in bark chip, a substrate that was novel to all calves. The walls were covered with black plastic to prevent calves from seeing the observers or calves in neighbouring pens. All testing took place between 10:00 and 14:30. Calves were moved from their home pen by two handlers to the arena where behaviour was recorded continuously in real-time at 30 frames/s for 20 min via digital handycams (SONY Handycam^®^ Camcorder DCR-SX65, Tokyo, Japan). The 20 min recording period started once the calf entered the arena and the gate was closed behind it. Calves were unfamiliar with the arena before the first test. Behaviours performed by experimental animals (calves that received saline or morphine) were scored continuously as durations (running) and frequencies (running, kicking, bucking, jumping, frontal pushing and mounting) (Table 1). One trained observer was used to analyse all video recordings; intra-observer reliability was calculated by scoring two arena videos and comparing agreement.

Lying behaviour in the home pen was recorded continuously using HOBO pendant G data loggers (64k, Onset Computer Corporation, Bourne, MA, USA), which have previously been validated for use in calves [20]. The loggers were fitted onto the calves hind legs on the first day of testing as described in Sutherland et al. [21] and lying behaviour was recorded at 60 s intervals, recording the Y and Z axes, from 2 h prior to testing in the arena test to 12 h post-testing. The data from the loggers were downloaded using Onset HOBOware Pro software (version 3.7.2, Onset Computer Corporation, Bourne, MA, USA) and raw g-force readings were converted into daily summaries of lying time in SAS 9.3 (SAS Institute Inc., Cary, NC, USA) using a code designed for this purpose [22].

#### 2.2.2. Cortisol Concentrations

Calves were blood sampled prior to and immediately after testing in the play arena. Blood samples (10 mL) were obtained by jugular venipuncture and collected into one additive-free evacuated tube (BD Vacutainer, Franklin Lakes, NJ, USA) to measure cortisol concentrations. Samples were held at ambient temperature following collection for up to 2 h to allow clotting and then centrifuged at 1500× *g* (3000 rpm) for 10 min at ambient temperature. Serum was then aspirated and aliquots were stored at −20 °C for future analysis. Cortisol concentrations were measured using a commercially available solid phase single antibody radioimmunoassay kit (Coat-a-Count^®^ Cortisol; Siemens; Los Angeles, CA, USA) by Gribbles Veterinary Pathology Ltd. (Hamilton, New Zealand).

### 2.3. Statistical Analysis

Data were analysed by ANOVA using GenStat 15th Edition (VSN International Ltd., 2013, Hemel Hempstead, UK). The data for all measures was normally distributed according to an inspection of residual plots and there was no evidence of skewness. Pen and treatment order within pen were fitted as random effects and treatment as the fixed effect. Lying behaviour was summarised into 4-h time periods for the 12 h immediately after treatment. Play data were summarised into 5-min periods over the 20 min of the test. These periods were analysed separately. A repeated measures analysis over time was also performed with treatment, time and the interaction as fixed effects and pen, treatment order within pen and time within treatment order within pen as random effects. Social play was infrequent, averaging 0.41 plays over 5 min, with a maximum of 3 plays. This data was analysed as binary data of the occurrence of any social play over the 5 min as a generalised linear mixed model with a logit link with fixed and random effects the same as for the repeated measures ANOVA. Fisher’s least significant differences test was used to detect any differences between and within treatments.

## 3. Results

The *F* and *p*-values from the statistical analysis for all variables are presented in Table 2.

### 3.1. Behaviour 

There was a significant treatment by time interaction for running/locomotor play (*F*(9,120) = 2.492, *p* = 0.012; Figure 1). During all time periods in the arena test, total time spent running/locomotor play did not differ (*p* > 0.05) between calves given saline or morphine, regardless of whether they were tested individually or in pairs. However, calves tested in pairs spent more (*p* < 0.05) time performing locomotor play during the first 5 min in the arena test. Overall, individually tested calves spent less time running/locomotor play than calves tested in pairs (running/locomotor play (s/5 min): IND: 22.8 ± 2.62; SOC: 32.3 ± 2.62, *F*(1,29) = 13.391, *p* = 0.001), but there was no difference between calves given saline or morphine (running/locomotor play (s/5 min): SAL: 27.5 ± 2.62; MOR: 27.6 ± 2.62, *F*(1,29) = 0.003, *p* = 0.956).

There was a tendency for the frequency of running/locomotor play bouts to be affected by treatment (*F*(3,29) = 2.730, *p* = 0.062), but there was no significant treatment by time interaction (*F*(9,120) = 1.265, *p* = 0.263; Figure 2). Overall, the frequency of running/locomotor play bouts did not differ between calves given saline or morphine (locomotor play bouts (no./5 min): SAL: 4.7 ± 0.35; MOR: 5.2 ± 0.35, *F*(1,29) = 2.625, *p* = 0.116), however, calves tested in pairs performed more locomotor play bouts compared with individually tested calves (locomotor play bouts (no./5 min): IND: 4.5 ± 0.35; SOC: 5.4 ± 0.35, *F*(1,29) = 5.506, *p* = 0.026).

The frequency of total individual play events was affected by treatment (*F*(3,29) = 3.101, *p* = 0.042; Figure 3), but there was no significant treatment by time interaction (*F*(9,120) = 1.563, *p* = 0.134). Over the entire 20 min arena test period, the frequency of total individual play events did not differ (*p* > 0.05) between calves given morphine or saline when tested individually or when tested in pairs, however, calves given morphine and tested in pairs performed fewer (*p* < 0.05) individual play events than calves given morphine and tested individually. Overall, individually tested calves performed more total individual play events than calves tested in pairs (total individual play events (no./5 min): IND: 5.3 ± 0.54; SOC: 3.7 ± 0.54, *F*(1,29) = 9.230, *p* = 0.005), but there was no difference between calves given saline or morphine (total individual play events (no./5 min): SAL: 4.6 ± 0.54; MOR: 4.4 ± 0.54, *F*(1,29) = 0.168, *p* = 0.685).

During the 20 min arena test period, calves given morphine tended to perform more social play events than calves given saline (*F*(1,10) = 2.854, *p* = 0.091; Figure 4). There was no significant treatment by time interaction (*F*(9,85) = 1.602, *p* = 0.590).

Individually tested calves that received morphine spent less (*p* < 0.05) time lying than individually tested calves that received saline during the first 4 h after returning to the home pen (*F*(3,29) = 4.91, *p* = 0.007; Figure 5). During the first 4 h after returning to the home pen, calves given morphine spent less time lying than calves given saline (lying (%/1–4 h post-testing): SAL: 52.4 ± 1.97; MOR: 42.7 ± 1.97, *F*(1,29) = 11.534, *p* = 0.002), but there was no difference in lying times between calves tested individually or in pairs (lying (%/1–4 h post-testing): IND: 45.7 ± 1.97; SOC: 49.4 ± 1.97, *F*(1,29) = 1.704, *p* = 0.202). There was no difference (*p* > 0.05) in lying times among treatments during the 5−8 or 9−12 h post-test periods.

### 3.2. Cortisol Concentrations

Cortisol concentrations were similar among all treatments prior to testing in the arena (*F*(3,29) = 0.500, *p* = 0.685; Figure 6). After exposure to the 20 min arena test, cortisol concentrations were higher in calves given saline than calves that received morphine regardless of whether they were tested individually or in pairs (*F*(3,29) = 7.121, *p* = 0.001; Figure 6).

## 4. Discussion

Administration of morphine tended to increase the performance of social play but had no effect on non-social play (e.g., locomotor play and individual play events) in calves in the present study. Several studies have investigated the relationship between opioids and social play in rats and have shown that the central opioid system is involved in modulating the expression of social play specifically through stimulation of the µ-opioid receptors (reviewed by Vanderschuren et al., [23]). Guard et al. [13] also found that morphine administration in marmosets increased social play, but similar to our findings, morphine administration had no effect on non-social play elements, such as locomotor play. Therefore, administration of a µ-opioid receptor agonist, such as morphine sulphate, may modulate play behaviour in calves in a similar mode as in rats and primates, however the effect of morphine on social play was rather small in this study and further research is needed to investigate other aspects of social play before this conclusion can be made.

It has been suggested that the performance of play behaviour may be a good indicator of positive emotions and good welfare in animals [7]. However, play consists of a complex repertoire of different behaviours and some aspects of play may be more linked to positive emotional states than others [24]. Indeed, social play is modulated in rats by the endogenous opioid system [23] and has been shown to be modulated by the same neural systems that also mediate the positive subjective and motivational properties of feeding and sexual behaviour [6]. The results of the present study and Guard et al. [13] suggest that non-social play (e.g., locomotor play, individual play events) is not modulated by the endogenous opioid system. Studies in humans also suggest that the different qualitative aspects of play may be important for wellbeing [24]. For example, maltreated children show more solitary than social play compared to non-maltreated children, even though total play behaviour was not altered [25]. Hospitalized [26] and depressed children [27] also show a higher proportion of solitary play compared to social play. Therefore, social play may be a better indicator of positive emotions than locomotor play. However, further evaluation of the relationship between positive affective state and different play elements in calves is needed. Furthermore, due to the low incidence of social play, test scenarios that motivate calves to perform more social play behaviours or give them more opportunity (e.g., larger space allowance, larger group sizes, increased testing duration) would be required to make this a more practical measure of positive emotions.

Calves given morphine prior to testing in the arena spent less time lying during the first 4 h after they had been returned to the home pen in the present study. A reduction in lying time indirectly infers that these calves may have been more active at this time than calves given saline. Similarly, morphine administration increased locomotor activity in rats [28], primates [13] and sheep [18]. Verbeek et al. [18] hypothesized that this increase in activity was related to increased goal directed or approach behaviour mediated by the dopaminergic pathways stimulated through the administration of morphine. However, locomotor activity (running/locomotor play) in the arena test did not appear to be increased in calves given morphine. In future studies, it would be of interest to evaluate general locomotor activity, separate from play behaviour, in calves given morphine to determine if morphine does cause an increase in general locomotor activity. 

Calves predominantly play (social and individual play such as kicking, bucking, and frontal pushing) while they are running, hence the terminology ‘locomotor play’ is often used when describing this behaviour. We would therefore expect that calves that spend more time running would also perform more play events. However, even though paired calves spent more time in running/locomotor play, they showed fewer total individual play events. The increase in locomotor play in the paired calves is possibly due to social facilitation, but it is not clear why this did not lead to more individual play events. Similarly, Jensen et al. [29] found that calves housed individually in large pens spent proportionally less time galloping when performing locomotor play than calves group housed in large pens, and subsequently performed more individual play events, such as leaping and bucking, than group housed calves. Jensen [30] also found that calves tested in pairs during an arena (open-field) test performed more locomotor play than calves tested individually, but did not record individual play events. Therefore, social facilitation may provoke calves to run more while isolation may stimulate calves to perform more play behaviours such as bucking and kicking. These results emphasise the importance of using the appropriate testing scenario for the behaviours that are of interest.

Peripheral administration of morphine suppressed cortisol release in response to exposure to a 20 min arena test in calves in the present study. Similarly, morphine suppressed cortisol release in response to milking in dairy cows [14] and reduced baseline cortisol concentrations in dairy cattle [31]. In dairy cattle, it has been shown that the HPA axis is under suppressive opioidergic control [31]. Therefore, our results confirm that the concentration of morphine given in the present study was sufficient to elicit a neuroendocrine response in calves and demonstrated that calves respond in a similar way to administration of morphine as adult cattle [14,31] and sheep [16]. Interestingly, cortisol concentrations were elevated in calves given saline after exposure to the arena test in the present study. Cortisol concentrations may have been elevated in saline calves as the novel arena may have elicited a fear response in these animals [32]. The suppressed cortisol response in the morphine-treated animals could be interpreted as a reduction in stress experienced by calves in the novel environment. However, studies in rodents show that morphine’s facilitating effects on social play do not depend on whether the play environment is novel or familiar, suggesting that morphine does not reduce fear of novel environments [33]. Activation of the HPA axis reflects mostly arousal and is not necessarily an indicator of a negative emotional state or poor welfare [34]. It is possible that morphine reduced the arousal aspect of emotional states during play, although this was not reflected in changes in locomotor behaviour.

Social play behaviour can also shape the early development of the HPA axis. It has been shown that marmosets who engage frequently in social play have lower cortisol responses to a stressor [35]. It has also been suggested that engaging in social play behaviour can reduce behavioural signs of distress [36]. Rats deprived of social play early in life show abnormal social behaviours and increased HPA axis responses to a social challenge later in life [36,37]. These findings highlight the importance of social play for normal development of the HPA axis and social behaviour and as a component of good animal welfare.

## 5. Conclusion

The administration of morphine tended to increase the performance of social play but had no effect on non-social (locomotor, individual) play in calves. Therefore, it appears as though administration of a µ-opioid receptor agonist may modulate play behaviour in calves using a similar mode of action as in rats and primates, although more research is needed to confirm this due to the small effects observed in this study.

## Figures and Tables

**Figure 1 animals-09-00056-f001:**
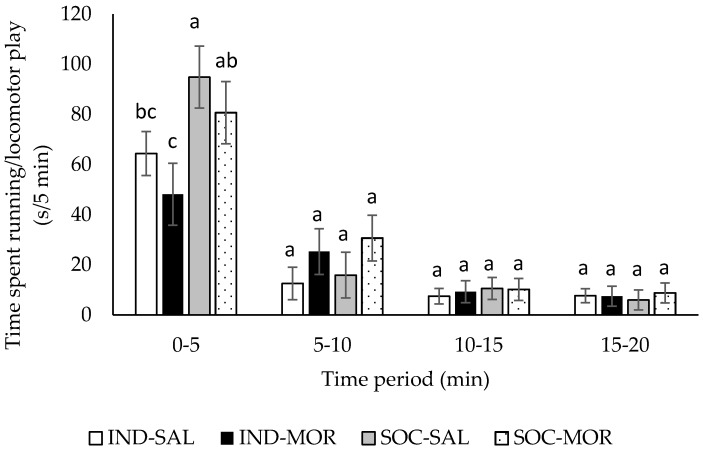
Total time calves spent running/locomotor play (s/5 min; Least square means ± SEM) during a 20 min observation period in an arena test. Calves received an injection of saline and were tested individually (IND-SAL, n = 12) or in pairs (SOC-SAL, n = 12), or received an injection of morphine and were tested individually (IND-MOR, n = 12) or in pairs (SOC-MOR, n = 12). For each time period, least square means with different superscripts differ at *p* < 0.05.

**Figure 2 animals-09-00056-f002:**
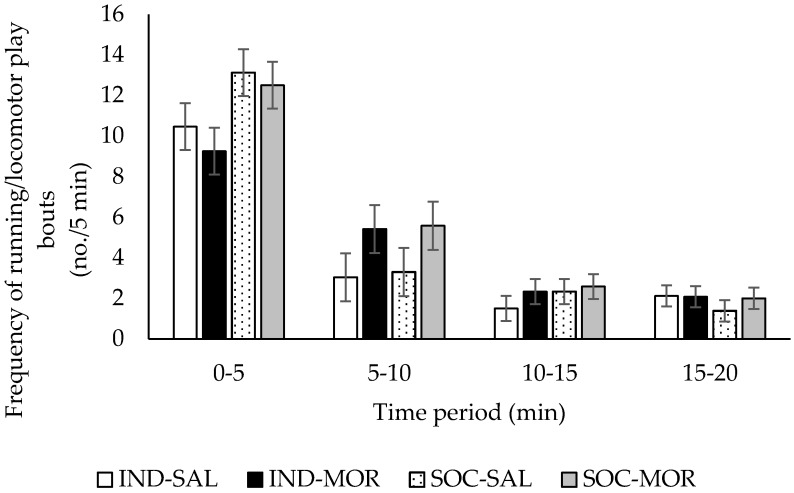
Frequency of running/locomotor play bouts (no./5 min; Least square means ± SEM) performed by calves during a 20 min observation period in an arena test. Calves received an injection of saline and were tested individually (IND-SAL, n = 12) or in pairs (SOC-SAL, n = 12), or received an injection of morphine and were tested individually (IND-MOR, n = 12) or in pairs (SOC-MOR, n = 12).

**Figure 3 animals-09-00056-f003:**
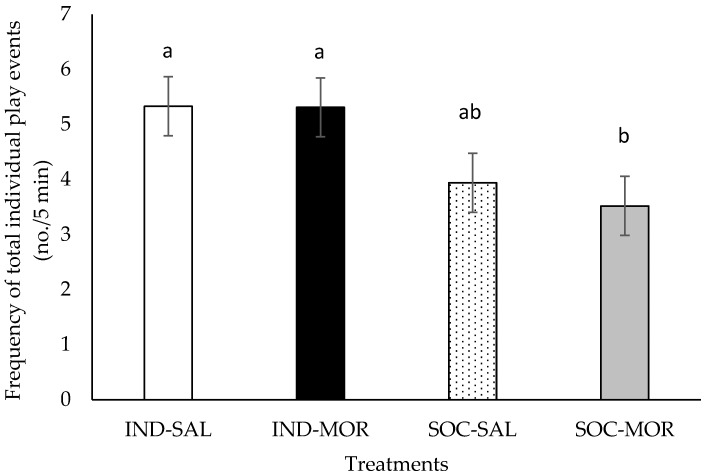
Frequency (no./5 min; Least square means ± SEM) of total individual play events performed by calves during a 20 min observation period in an arena test. Calves received an injection of saline and were tested individually (IND-SAL, n = 12) or in pairs (SOC-SAL, n = 12), or received an injection of morphine and were tested individually (IND-MOR, n = 12) or in pairs (SOC-MOR, n = 12). Least square means with different superscripts differ at *p* ≤ 0.05.

**Figure 4 animals-09-00056-f004:**
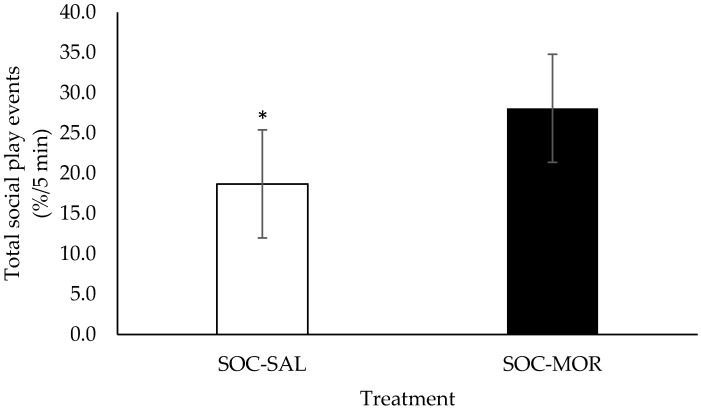
Total social play events (%/5 min; Least square means ± SEM) performed by calves during a 20 min observation period in an arena test. Calves received an injection of saline (SOC-SAL, n = 12) or received an injection of morphine (SOC-MOR, n = 12). * Least square means differ at *p* < 0.10.

**Figure 5 animals-09-00056-f005:**
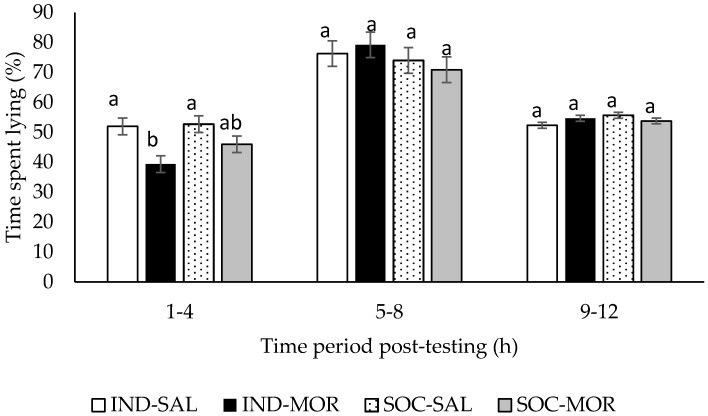
Time (%; Least square means ± SEM) calves spent lying in the home pen after the arena test. Calves received an injection of saline and were tested individually (IND-SAL, n = 12) or in pairs (SOC-SAL, n = 12), or received an injection of morphine and were tested individually (IND-MOR, n = 12) or in pairs (SOC-MOR, n = 12). For each time category, least square means with different superscripts differ at *p* < 0.05.

**Figure 6 animals-09-00056-f006:**
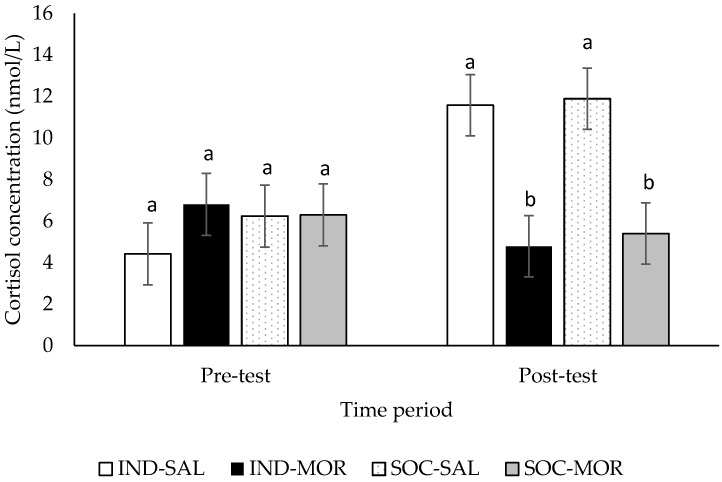
Cortisol concentrations (nmol/L; Least square means ± SEM) of calves prior to and immediately after being tested in a 20 min arena test. Calves received an injection of saline and were tested individually (IND-SAL, n = 12) or in pairs (SOC-SAL, n = 12), or received an injection of morphine and were tested individually (IND-MOR, n = 12) or in pairs (SOC-MOR, n = 121). For each time category, least square means with different superscripts differ at *p* < 0.05.

**Table 1 animals-09-00056-t001:** Description of behaviours.

Behaviour	Description
Running/locomotor play ^1^	Rapid forward or sideways movement, including trotting (two-beat gait), cantering (three-beat gait) and galloping (fast four-beat gait). Lasting longer than 1 s in real time.
*Individual Play Events*	
Kicking ^2^	One or both hind legs lifted off the ground in one rapid movement, legs extended outwards from the body. The calf can be stationary or moving.
Bucking ^2^	One or both hind legs lifted off the ground in one rapid movement and extended outwards from the body. Hooves are raised as high as or higher than the front knees of the forelegs.
Jumping ^1^	Both forelegs lifted from the ground and stretched forward. Movement upwards but not forwards. The hind legs may be elevated.
*Social Play Events*	
Frontal pushing ^1^	Two calves are standing front to front, butting head against head/neck.
Mounting ^1^	A calf mounts another calf’s head or body from front, side or back.

^1^ Definitions based on the ethogram described by Worth et al., [19]; ^2^ Definitions were based on the ethogram described by Sutherland et al., [17].

**Table 2 animals-09-00056-t002:** *F* and *p*-values from statistical analysis for all measured variables. Calves received either an injection of saline (SAL) or morphine (MOR) and were tested either individually (IND) or in pairs (SOC) in an arena test.

Variable	Treatment		SAL vs. MOR		IND vs. SOC		Time		Treatment × Time	
	*F*_(3,29)_-Value	*p*-Value	*F*_(1,29)_-Value	*p*-Value	*F*_(1,29)_-Value	*p*-Value	*F*_(3,120)_-Value	*p*-Value	*F*_(9,120)_-Value	*p*-Value
Duration of running/locomotor play	4.266	0.013	0.003	0.956	13.391	0.001	5.781	0.001	2.492	0.012
Frequency of running/locomotor play	2.730	0.062	2.625	0.116	5.506	0.026	5.781	0.001	1.265	0.263
Frequency of total individual play events	3.101	0.042	0.168	0.685	9.230	0.005	5.781	0.001	1.563	0.134
Percentage of total social play events	2.854	0.091	-	-	-	-	1.602	0.195	0.830	0.590
Time spent lying (1–4 h post-treatment)	4.914	0.007	11.534	0.002	1.704	0.202	-	-	-	-
Time spent lying (5–8 h post-treatment)	0.646	0.592	0.000	0.985	1.464	0.236	-	-	-	-
Time spent lying (9–12 post-treatment)	2.041	0.130	0.050	0.825	1.558	0.222	-	-	-	-
Cortisol concentrations (pre-test)	0.500	0.685	0.678	0.417	0.200	0.658	-	-	-	-
Cortisol concentrations (post-test)	7.121	0.001	13.391	0.001	0.104	0.749	-	-	-	-

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
