# Peer review of "Effect of Morphine Administration on Social and Non-Social Play Behaviour in Calves"

_animals, 2019, doi:10.3390/ani9020056_

Round 1
Reviewer 1 Report
This is a well-designed study addressing an interesting and important question. The paper is well organized and easy to read.
Nevertheless, I have a few comments on the statistics and the interpretation of the results as well as some minor suggestions for modifications.
1. My major concern is about the statistics of the main result, ie. the effect of morphine on social play. There are several issues with it:
1.a. The authors state on lines 151-152 that the no transformations were needed as the normality assumptions were met. However, looking at the y axis of Fig 4, the average frequency of social play events was 0.25 and 0.5 per 5 min for the two treatments, respectively. It follows logically from these mean values that in more than half of the measured 5 min periods, the number of social play events was 0. Thus clearly the data for social play were NOT normally distributed and the deviation from normality was not mild or trivial. For data with such distribution (most of the data equal to 0), it is not correct to apply standard linear mixed model. Such data are also impossible to transform into normal distribution. One possibility would be to employ a generalised linear mixed model, either with binary data (Did social play occur within the 5 min? Yes or No) with the logit link function, or possibly a Poisson distribution with log link function.
1.b. Based on section 2.3, I assume that the model of social play frequency was structured similarly as the other models, except that only paired testing was included. If so, then the treatment by time interaction should be mentioned in the results for social play, whether it was significant or not.
1.c. Strictly speaking, the result is (in statistical jargon) only a “tendency” as the p value is P= 0.051 (line 196) and therefore 0.05 < P < 0.10. Given the fact that a different model will need to be calculated for the social play data (see point 1.a.), it remains to be seen whether there will be sufficient support for the main claim of the study.
2. The presentation of the statistical results is inconsistent it the text. Sometimes, the full information is given about the F value, degrees of freedom and the exact P value (e.g., lines 155-156, 161 etc.) but in other cases only the information whether P was below or above 0.05 is provided (lines 157, 158, 182). Full information should be provided for all the results.
3. I am not sure what the abbreviation “± SED” means. The two usual ways how to provide info about variation in the data is standard deviation (SD) or standard error (of the mean, SE). Is it SD or SE that is depicted in the graphs? SED could possibly also mean “standard error of the difference (between means)” but this is a quantity that relates to a difference between two mean values and therefore should not appropriately be linked with a least square mean of the treatment level.
4. More specifically to the assumptions of the mixed models (lines 151-152): were the residuals tested for normal distribution? Was homoscedasticity tested?
5. Comparing Figures 2, 3 and 4, it becomes clear that social play contributed with only 5% to the total frequency of play (about 0.4 events / 5 minutes) while the remaining 95% of play were either the locomotor ( about 5 events / 5 minutes) or the individual (4 events / 5 minutes) play. This fact should be mentioned in the discussion because the scarcity of social play affects the practicability of using it as indicator of positive emotions in calves. This fact is linked to point 1.a.: in most of the 5 min intervals, the calves performed no social play at all.
6. The logic of concluding that social play may be a better indicator of positive emotions than locomotor play is not straightforward. Positive emotions were not independently measured in this study so a direct inference from the current data about emotions cannot be made. Social play is modulated by a number of neurotransmitter systems (see the cited review of Vanderschuren et al 2016 NBR) and it would be an oversimplification to say that any behaviour that is modulated by the opioid system indicates positive emotions whereas any behaviour that is not modulated by opioids has nothing to do with positive emotions. Also, the distinction between social and non-social behaviour is not as straightforward as it may seem. For instance, the study of Guard et al on marmosets classified “approach withdraw play” among social play, although there was no physical contact between the playing individuals, just a co-ordinated or oriented locomotor movement. A similar behaviour (e.g., running or walking towards another calf, or running behind the play partner) was probably classified as non-social in this study. The transition between solitary locomotor play – synchronized / parallel locomotor play – mutually oriented locomotor play – social play is more fluid than it seems from the traditional play categorization into locomotor, social and object play. These comments are not meant to dismiss the results of the study. Rather, I encourage the authors to make the discussion more reserved and nuanced.
Line 178 replace “effected” by “affected”
Line 248 replace “maybe” by “may be”
Line 265 replace “calves housed individually proportionality spent more time performing individual play” by “calves housed individually spent proportionally more time performing individual play”
In Fig 4, use SOC-SAL and SOC-MOR for description of the Treatments
In Fig 5, the superscript “a” is missing above one of the bars.
In conclusion, I recommend a revision of the paper, especially based on a correct statistical model for the effects social play, as outlined in points 1.a.-1.c.
Author Response
This is a well-designed study addressing an interesting and important question. The paper is well organized and easy to read.
Au: Thank you.
Nevertheless, I have a few comments on the statistics and the interpretation of the results as well as some minor suggestions for modifications.
1. My major concern is about the statistics of the main result, ie. the effect of morphine on social play. There are several issues with it:
1.a. The authors state on lines 151-152 that the no transformations were needed as the normality assumptions were met. However, looking at the y axis of Fig 4, the average frequency of social play events was 0.25 and 0.5 per 5 min for the two treatments, respectively. It follows logically from these mean values that in more than half of the measured 5 min periods, the number of social play events was 0. Thus clearly the data for social play were NOT normally distributed and the deviation from normality was not mild or trivial. For data with such distribution (most of the data equal to 0), it is not correct to apply standard linear mixed model. Such data are also impossible to transform into normal distribution. One possibility would be to employ a generalised linear mixed model, either with binary data (Did social play occur within the 5 min? Yes or No) with the logit link function, or possibly a Poisson distribution with log link function.
Au: The reviewer makes a valid point about the social play analysis. There were a lot of zeroes in the data, however, the residual plot looked acceptable. We have re-analysed the data though using the methodology suggested by the reviewer (a generalised linear mixed model with binary data (did social play occur or not within the 5 minutes?). The statistical analysis and results section as well as figure 4 have been revised to reflect this new analysis.
1.b. Based on section 2.3, I assume that the model of social play frequency was structured similarly as the other models, except that only paired testing was included. If so, then the treatment by time interaction should be mentioned in the results for social play, whether it was significant or not.
Au: Yes, the model was structured in a similar way. This information has now been added.
1.c. Strictly speaking, the result is (in statistical jargon) only a “tendency” as the p value is P= 0.051 (line 196) and therefore 0.05 < P < 0.10. Given the fact that a different model will need to be calculated for the social play data (see point 1.a.), it remains to be seen whether there will be sufficient support for the main claim of the study.
Au: After the new analysis the P-value is greater, but still a tendency. The manuscript has been updated appropriately.
2. The presentation of the statistical results is inconsistent it the text. Sometimes, the full information is given about the F value, degrees of freedom and the exact P value (e.g., lines 155-156, 161 etc.) but in other cases only the information whether P was below or above 0.05 is provided (lines 157, 158, 182). Full information should be provided for all the results.
Au: Full information has now been provided for all results in the text and/or table 2. We have provided P-values along with F statistics and degrees of freedom when a hypothesis test was performed, such as the tests of treatment, time and treatment by time effects in the models used. Where post hoc tests were performed using Fisher’s protected Least Significant Differences (LSD) test, means that differed significantly have been indicated in the manuscript by P < or > 0.05. Fisher’s LSD does not correct for multiple comparisons and therefore exact P-values cannot be provided.
3. I am not sure what the abbreviation “± SED” means. The two usual ways how to provide info about variation in the data is standard deviation (SD) or standard error (of the mean, SE). Is it SD or SE that is depicted in the graphs? SED could possibly also mean “standard error of the difference (between means)” but this is a quantity that relates to a difference between two mean values and therefore should not appropriately be linked with a least square mean of the treatment level.
Au: Measures of variance have been changed to SEM throughout the results and the figures.
4. More specifically to the assumptions of the mixed models (lines 151-152): were the residuals tested for normal distribution? Was homoscedasticity tested?
Au: The residuals of the other variables were all normally distributed according to an inspection of residual plots and there was no evidence of skewness for these variables. This information has been added to the statistical analysis section.
5. Comparing Figures 2, 3 and 4, it becomes clear that social play contributed with only 5% to the total frequency of play (about 0.4 events / 5 minutes) while the remaining 95% of play were either the locomotor ( about 5 events / 5 minutes) or the individual (4 events / 5 minutes) play. This fact should be mentioned in the discussion because the scarcity of social play affects the practicability of using it as indicator of positive emotions in calves. This fact is linked to point 1.a.: in most of the 5 min intervals, the calves performed no social play at all.
Au: This is a very good point and has been added to the discussion.
6. The logic of concluding that social play may be a better indicator of positive emotions than locomotor play is not straightforward. Positive emotions were not independently measured in this study so a direct inference from the current data about emotions cannot be made. Social play is modulated by a number of neurotransmitter systems (see the cited review of Vanderschuren et al 2016 NBR) and it would be an oversimplification to say that any behaviour that is modulated by the opioid system indicates positive emotions whereas any behaviour that is not modulated by opioids has nothing to do with positive emotions. Also, the distinction between social and non-social behaviour is not as straightforward as it may seem. For instance, the study of Guard et al on marmosets classified “approach withdraw play” among social play, although there was no physical contact between the playing individuals, just a co-ordinated or oriented locomotor movement. A similar behaviour (e.g., running or walking towards another calf, or running behind the play partner) was probably classified as non-social in this study. The transition between solitary locomotor play – synchronized / parallel locomotor play – mutually oriented locomotor play – social play is more fluid than it seems from the traditional play categorization into locomotor, social and object play. These comments are not meant to dismiss the results of the study. Rather, I encourage the authors to make the discussion more reserved and nuanced.
Au: We definitely agree that more research is needed to support our findings and that further evaluation of the relationship between positive affective state and different play elements in calves is needed before any conclusions can be made. Therefore, we have tried to make the discussion more reserved.
Line 178 replace “effected” by “affected”
Au: Corrected.
Line 248 replace “maybe” by “may be”
Au: Corrected.
Line 265 replace “calves housed individually proportionality spent more time performing individual play” by “calves housed individually spent proportionally more time performing individual play”
Au: This sentence has been revised.
In Fig 4, use SOC-SAL and SOC-MOR for description of the Treatments
Au: Figure 4 has been revised and the figure legends changed.
In Fig 5, the superscript “a” is missing above one of the bars.
Au: The missing subscript has been added.
In conclusion, I recommend a revision of the paper, especially based on a correct statistical model for the effects social play, as outlined in points 1.a.-1.c.
Au: The paper has been revised taking into account the reviewers comments and the re-analysis of the social play behaviour.
Reviewer 2 Report
In the present study, the effect of morphine on play behavior in calves was investigated. Studies in rodents and primates have implicated the opioid system in play behavior, and the present study investigates this in ruminants. Of note, the calves were tested both individually and in pairs, and different categories of play behavior (i.e. locomotor, individual and social) were distinguished. In addition, activity of the hypothalamus-pituitary-adrenal axis was assessed, by measurement of cortisol concentrations in blood. The findings show that testing in pairs affects locomotor and individual play (increasing the former and reducing the latter), but morphine does not alter these behaviors. Morphine did, however, increase social play. Moreover, treatment with morphine reduced post-test lying in the pen, and reduced testing-induced cortisol concentrations. This is a valuable study, that not only extends the stimulatory effects of morphine on social play to another mammalian species, but also increases our understanding of the modulation of different forms of play behavior. I have a few things for the authors to consider.
Major points
1. I really like the distinction between locomotor, individual and social play behaviors made in this study, and I understand that these distinctions are based on previously published ethograms. That said, to me the ‘individual’ play behaviors seem locomotor in nature, so the distinction between the two feels a bit artificial (interestingly though, paired testing has differential effects on locomotor vs individual play, suggesting that these actually are different behaviors). Can the authors comment on this, and provide a more detailed justification for the distinction?
2. Testing took place according to a 2 x 2 design (individual vs social and saline vs morphine), and the 20 min test period was subdivided into 4 periods of 5 min each. Therefore, the data (locomotor play, individual play, lying) should be analyzed using a 2-way repeated measures ANOVA, whereby test condition, treatment and time are within-subjects factors (for the social play data, a 1-way repeated measures ANOVA should be used, and for cortisol, a 2-way ANOVA). Perhaps this is how the data were actually analyzed, but since not all statistical results are reported, this is hard to tell. I would advise the authors to analyze the data this way, report relevant significant/non-significant results in the text, and report the entire statistical analysis in a table.
Minor points
3. The statement that ‘endogenous opioids have been connected with reward processing independent of dopamine and μ receptor stimulation’ (line 48-49) is incorrect, inasmuch as that stimulation of the μ receptor is the primary mechanism by which opioids enhance reward processes.
4. Presentation of statistical results
-Fig. 1: What does the ‘b’ symbol above the SOC-MOR bar (0-5 min) signify? If the two SOC treatments differ from the two IND treatments, but not from one another, then it should suffice to label the two IND bars with a ‘b’.
-Fig. 5: The symbol above the IND-MOR 5-8 bar is missing.
5. Typo:
Abstract, line 30: ‘supressed’
Author Response
In the present study, the effect of morphine on play behavior in calves was investigated. Studies in rodents and primates have implicated the opioid system in play behavior, and the present study investigates this in ruminants. Of note, the calves were tested both individually and in pairs, and different categories of play behavior (i.e. locomotor, individual and social) were distinguished. In addition, activity of the hypothalamus-pituitary-adrenal axis was assessed, by measurement of cortisol concentrations in blood. The findings show that testing in pairs affects locomotor and individual play (increasing the former and reducing the latter), but morphine does not alter these behaviors. Morphine did, however, increase social play. Moreover, treatment with morphine reduced post-test lying in the pen, and reduced testing-induced cortisol concentrations. This is a valuable study, that not only extends the stimulatory effects of morphine on social play to another mammalian species, but also increases our understanding of the modulation of different forms of play behavior. I have a few things for the authors to consider.
Au: Thank you.
Major points
1. I really like the distinction between locomotor, individual and social play behaviors made in this study, and I understand that these distinctions are based on previously published ethograms. That said, to me the ‘individual’ play behaviors seem locomotor in nature, so the distinction between the two feels a bit artificial (interestingly though, paired testing has differential effects on locomotor vs individual play, suggesting that these actually are different behaviors). Can the authors comment on this, and provide a more detailed justification for the distinction?
Au: Yes, there is definitely an overlap as calves predominantly (over 80% of the time) perform individual or social play events while they were running. The distinction between these behaviours and what it might mean has now be clarified and elaborated on in the discussion.
2. Testing took place according to a 2 x 2 design (individual vs social and saline vs morphine), and the 20 min test period was subdivided into 4 periods of 5 min each. Therefore, the data (locomotor play, individual play, lying) should be analyzed using a 2-way repeated measures ANOVA, whereby test condition, treatment and time are within-subjects factors (for the social play data, a 1-way repeated measures ANOVA should be used, and for cortisol, a 2-way ANOVA). Perhaps this is how the data were actually analyzed, but since not all statistical results are reported, this is hard to tell. I would advise the authors to analyze the data this way, report relevant significant/non-significant results in the text, and report the entire statistical analysis in a table.
Au: The statistical analysis section of the manuscript has been revised to make it more clear what analysis was performed. Relevant significant/non-significant results have been left in the text, and the entire statistical analysis is now reported in table 2.
Minor points
3. The statement that ‘endogenous opioids have been connected with reward processing independent of dopamine and μ receptor stimulation’ (line 48-49) is incorrect, inasmuch as that stimulation of the μ receptor is the primary mechanism by which opioids enhance reward processes.
Au: Thank you for the clarification. This sentence has been revised to reflect the reviewers comments.
4. Presentation of statistical results
-Fig. 1: What does the ‘b’ symbol above the SOC-MOR bar (0-5 min) signify? If the two SOC treatments differ from the two IND treatments, but not from one another, then it should suffice to label the two IND bars with a ‘b’.
Au: Thank you for finding that error, the subscripts have not been revised.
-Fig. 5: The symbol above the IND-MOR 5-8 bar is missing.
Au: This has been corrected.
5. Typo:
Abstract, line 30: ‘supressed’
Au: Corrected
Round 2
Reviewer 1 Report
I am satisfied with the changes made to the manuscript. I my view, the paper is now fit for publication in Animals.